# Nano Catalysis of Biofuels and Biochemicals from *Cotinus coggygria* Scop. Wood for Bio-Oil Raw Material

**DOI:** 10.3390/polym14214610

**Published:** 2022-10-30

**Authors:** Xiaochen Yue, Guanyan Li, Xiangmeng Chen, Zhaolin Li, Haiping Gu, Huiling Chen, Wanxi Peng

**Affiliations:** 1School of Forestry, Henan Agricultural University, Zhengzhou 450002, China; 2Ecology and Nature Conservation Institute, Chinese Academy of Forestry, Beijing 100091, China

**Keywords:** *Cotinus coggygria* scop., TG-FTIR, pyrolysis, nano Mo/Fe_2_O_3_-catalysis, biochemistry

## Abstract

*Cotinus coggygria* Scop. as a precious landscape shrub and a good afforestation species that is used in the pharmaceutical industry. In this paper, TG-FTIR, TG-DTG, and Py-GC/MS were used to study the biomaterials of *Cotinus coggygria* used as biofuels and biochemicals under the catalysis of nano-Mo/Fe_2_O_3_. The wood powder was extracted using a methanol/benzene solution, and the extract was analyzed by FTIR and GC-MS. The results showed that the pyrolysis products of *Cotinus coggygria* wood were rich in phenols, alcohols, and biofuels. The metal nano-Mo powder played a catalytic role in the interpretation of the gas in the species, where it accelerates gas products. Metal nano-Fe_2_O_3_ has a certain flame-retardant effect on the burning process of *Cotinus coggygria* wood, and the residual amount of pyrolysis is greater. The contents of the extract Formamide, 1-Hexanol, Levodopa, and 9,12-Octadecadienoic acid (Z,Z)- are not only widely used industrially but also play an important role in medicine. *Cotinus coggygria* is therefore an excellent biomaterial for biofuels and biochemicals.

## 1. Introduction

*Cotinus coggygria* Scop. (CCS) was a deciduous tree and shrub of the lacquer tree family that was mainly distributed in southern Europe, Iran, Pakistan, northern India, southwestern China, North China, and Zhejiang. It is typically used as a shelter forest for water and soil conservation and for landscaping [1,2,3,4]. In addition, CCS contains various chemical components, such as flavonoids and gallic acid, and is also a traditional medicinal plant in China. The CCS leaf extract had a significant anti-inflammatory effect. The Beijing University of Traditional Chinese Medicine used CCS leaves to treat more than 500 cases of cervical erosion and achieved satisfactory results. The leaves were also used to treat 400 cases of acute jaundice hepatitis, with a cure rate of 93%. The CCS root bark can heal women’s postpartum strain [5,6,7].

CCS is an important landscape and medicinal plant that has been extensively studied in recent years [8,9]. CCS leaves and flowers can be distilled to obtain a variety of volatile oil components. Antal et al. (2010) used the HPLC method to identify and analyze 14 phenolic substances in CCS wood extracts. The concentration of two new natural compounds C-3/C-3″ keto diocytes was 0.03–0.06% [10]. Miao et al. (2017) reported that six phenolic acids were isolated from the leaves of CCS [3]. Marcetic et al. (2014) screened CCS leaves in Bulgaria to investigate the in vitro antioxidant activity of water extracts; the results showed that they had good antioxidant activity [11]. Totada used a CCS aqueous extract in combination with insulin, and studies suggest that combination therapy can promote eyelash growth and hair growth [12]. Ge et al. (2022) studied the potential value of *Pinus armandii* by using TG, PY-GC-MS, FTIR, NMR, and other technologies, and found that *Pinus armandii* was rich in natural product active molecules and bio-oil, which can be used for the development and utilization of the biomedical industry [13]. Ma et al. (2022) used FTIR, GCMS, LC-QTOF-MS, and other methods to analyze the bioactive components of *Ginkgo biloba* branches and realized the zero waste multi-level utilization of biomass of *Ginkgo biloba* branches [14]. The scavenging capacity of *Ginkgo biloba* branches’ bioactive compounds to free radicals reached 98.18%. The mechanism of action may be that CCS extract induces the hair growth factor, thus regulating hair growth and the growth cycle. In a study of anticoagulation and blood pressure reduction, Enxian and others have shown that the CCS rhizome aqueous extract can significantly shorten the length of thrombus, reduce the wet weight and dry weight of thrombus, reduce the thrombus index, prolong thrombin time, and entericulate with aspirin [15]. Compared to standard treatment, there was no significant difference, and the drug can be considered to have obvious anticoagulant and antithrombotic effects. In addition, Olmez et al. studied CCS syrup and found that CCS had obvious protective and repairing effects on liver damage caused by hepatitis, and can significantly improve immune cell activity and immune organ function [16].

Based on previous research, this paper mainly focuses on CCS wood and its organic solvent extract, and added nano-metal catalyst to wood powder for comparison [17,18,19]. TG-DTG, TG-FTIR, and Py-GC/MS methods were used to analyze CCS wood as seen in Figure 1. The CCS wood extracts were tested by GC-MS and FTIR methods to investigate the thermal decomposition characteristics and chemical composition, to develop CCS into a high-quality resource, and to provide a theoretical reference for future research. 

## 2. Materials and Methods

### 2.1. Experimental Materials

The wood of CCS was obtained from the Xixia Forest Farm in Nanyang City, Henan Province. The wood samples were extracted with methanol/benzene (1:1) and named A1. The amount of organic solvent was 300 mL, the extraction time was 4 h, and the extraction temperature was 50 °C. Finally, a rotary evaporator was used to concentrate the solution to 30 mL to obtain the extracts.

MACKLIN Nano-Mo (80 nm) with a purity of 99.9% and MACKLIN nano-Fe_2_O_3_ (30 nm) with a purity of 99.5% were prepared by Shanghai Macklin Biological Co., Ltd. in Shanghai, China. The sample was dried, broken into 20 g pieces, and divided into four 5-g portions as seen in Table 1. The first 5-g bark piece was called B1; the second 5-g bark piece was mixed with 0.05 g nano-Mo and called B2; the third 5-g bark piece was mixed with 0.05 g nano-Fe_2_O_3_ and named B3; and the fourth 5-g bark piece was mixed with 0.025 g Mo and 0.025 g Fe_2_O_3_ and called B4.

### 2.2. Experimental Methods

Thermogravimetric Analysis (TG-DTG) experiments

We performed TG-DTG analysis on the powder of the sample using a thermogravimetric analyzer (TGA-Q500, TA Instruments, NewCastle, DE, USA). The temperature program for TG started at 30 °C and increased to 850 °C at heating rates of 25 °C/min and 55 °C/min [20,21], respectively.

Thermogravimetric–Fourier-Transform-Infrared (TG-FTIR) experiments

The samples of B1, B2, B3, and B4 were analyzed using a thermogravimetric analyzer connected to a Fourier-transform infrared spectrometer. The temperature was increased from 50 °C to 850 °C at a rate of 55 °C/min. The IR spectra were recorded at 4000–400 cm^−1^ with a resolution of 1 cm^−1^. Three-dimensional (3D) FT-IR spectrograms were obtained after the experiments with a Thermo Scientific Nicolet™ 6700 FT-IR spectrometer [22,23]. 

Pyrolysis-gas chromatography-mass (Py-GC/MS) analysis

We used pyrolysis-gas chromatography-mass spectrum (CDS 5000-Agilent 7890B-5977A; Agilent Technologies, Palo Alto, CA, USA) analysis sample powders B1, B2, B3, and B4. The thermal solution temperature was 850 °C, the temperature increase rate was 20 °C/ms, and the heat solution time was 15 s [24,25]. The initial temperature of the GC program was 40 °C, the temperature increased at 8 °C/min to 120 °C, and then increased at a heating rate of 10 °C/minute to 200 °C for 5 min.

FT-IR analysis

The infrared spectrum data of the extracted samples were obtained using an FT-IR spectrophotometer (iS10, Thermo Fisher Scientific, Waltham, MA, USA).

GC-MS analysis

GC/MS determination: The samples were performed on a gas chromatography-mass spectrum (Agilent 7890B-5977A; Agilent Technologies, Palo Alto, CA, USA). The column was HP-5MS (250 μm × 0.25 μm × 30 m). High purity helium was used as the carrier gas, the flow rate was 1 mL/min, and the split ratio was set to 5:1. The temperature program for GC started at 50 °C, increased to 250 °C at a rate of 8 °C/min, and then increased to 280 °C at a rate of 5 °C/min. The following variables were used: Scan mass range program of 35–580 amu.

## 3. Results and Discussion

Behavior during Combustion of CCS wood

### 3.1. TG-DTG Analysis

The burning characteristics and moisture content of wood determine whether it has excellent flame retardancy, and also determines the broad application prospects of its industrialization [26]. For this reason, thermal stability analysis is an effective method for assessing wood flame retardancy and the prospect for industrial applications [27,28,29]. Figure 2 and Figure 3 show the weight loss of CCS wood powder at a heating rate of 25 °C/min and 55 °C/min, respectively. As shown in Figure 2, the weights of the four sample wood powders, B1, B2, B3, and B4, decreased from 100% to 2.98%, 9.51%, 15.83%, and 15.18%, respectively. Overall, the curve decline can be divided into three phases. The first was at 30–200 °C, which was a micro-weight loss process, mainly due to the evaporation of water and the removal of a small amount of volatile compounds [30,31]. In this process, the weight loss rates of the four groups of samples were 8.54%, 7.97%, 6.76%, and 6.03%, respectively. The DTG curve was essentially zero at this stage without significant changes. The second stage was from 200 °C to 400 °C. This process was the main loss stage of the weight; the organic component material was severely cracked in the process, and the cellulose, hemicellulose, and lignin were decomposed. At this stage, the weight loss rates of the four groups of samples were 85.56%, 79.80%, 74.07%, and 74.76%. The DTG curve gradually increased from around 200 °C and peaked at around 370 °C. The last stage was at 400–850 °C. At this stage, the weight declined less and gradually reached a plateau. This stage was mainly a subsequent decomposition reaction of several organic components as the temperature continued to increase. The DTG curve began to decrease gradually at this stage and eventually tended to zero. A comparison of the four weight loss curves indicated that the weight loss rate of B1 samples without nano-metal catalyst was the highest, the weight loss rate of sample B2 with nano-Mo was second, and the weight loss rates of the B3 and B4 samples were basically identical, and less than B2. These results show that the nano-metal Mo and Fe_2_O_3_ do not play a significant role in catalyzing the thermal decomposition of CCS wood powder, but play a specific role in flame retardant pyrolysis [32,33]. Among these, the flame-retardant effect of nano-Fe_2_O_3_ was more significant.

However, Figure 3 shows that the weight loss rates of the final four samples were 86.44%, 85.41%, 86.23%, and 85.07%, respectively. The comparison of Figure 2 and Figure 3 shows that the flame-retardant effect of nano-metals changed when the heating rate increased to 55 °C/min. The final residual amount of wood powder was basically unchanged and showed no apparent difference. These results showed that an increase in the heating rate can inhibit the flame-retardant characteristics of nano-metals on CCS wood powder. This experiment shows that in the industrial production and utilization of CCS, if the scutellaria are stabilized and not decomposed, the temperature should be controlled below 200 °C as much as possible. At the same time, a small amount of nano-metal material can be added to prevent resistance. However, with regard to the combustion effect, if CCS wood powder thermal decomposition processing is to be utilized, it is possible to suppress part of the flame-retardant effect by increasing the heating rate.

### 3.2. TG-FTIR Analysis

The structure of the 3D-FTIR spectrum of the pyrolysis volatiles is shown in Figure 4 for the B1, B2, B3, and B4 samples. TG-FTIR technology analyzes the mass change characteristics of the sample pyrolysis gasification process and quickly analyzes the formation and characteristics of the release of gaseous products [34]. Figure 5 shows the cleavage of chemical bonds and the production of other small molecules for each sample during heating. During the initial stage of pyrolysis, the characteristic peak is at 3800–3500 cm^−1^, mainly indicating the cleavage of the O–H bond and the generation of H_2_O gas. This process was mainly caused by the evaporation of residual moisture in the sample. The IR light absorption characteristic frequency of the CO_2_ gaseous product is 2400–2200 cm^−1^, the IR light absorption characteristic frequency of CH_4_ is 3200–2700 cm^−1^, and the IR light absorption characteristic frequency of the carbonyl group (C=O) is 1900–1650 cm^−1^. At the same time, strong absorption peaks appear in the range of 3000–2650 cm^−1^, 1850–1600 cm^−1^, and 1500–1000 cm^−1^, indicating stretching vibration of the C–H bond, C=O double bond stretching vibration, and C–H bond bending vibrations in the plane, respectively [35,36]. These absorption peaks correspond to macromolecules, such as ketones, aldehydes, alcohols, and acids. As the temperature continues to increase, the thermal decomposition stage of the pyrolysis residue finally occurs, at which time the C=O precipitates as the main gas product. The C–H and C=O bonds are further cleaved and converted by aromatization.

Based on the above analysis, the pyrolysis gasification process of bio-oil could mainly be divided into two stages: volatilization pyrolysis of the light components under low temperature and pyrolysis gasification and condensation coking of the heavy components under high temperature. A comparison of the results of the four groups of samples showed that B2 and B4 were released after the addition of the nano-Mo powder. For infrared detection, the absorbance was significantly higher than in B1, which indicated that Mo powder catalyzed the release of pyrolysis products, H_2_O, and CO_2_ gases. The content was higher than the gas released by B1. The addition of Fe_2_O_3_ to B3 had no significant difference in gas absorbance compared to B1, indicating that Fe_2_O_3_ has no obvious catalytic effect on the release of gaseous products during pyrolysis. In addition, the size of the metal nanoparticles largely affects their catalytic function, and future TG-FTIR studies can begin by changing the type of catalyst and reducing the particle size of the catalyst.

### 3.3. Py-GC/MS Analysis

The total ion flow maps of the samples from Py-GC/MS were shown in Figure 6, Figure 7, Figure 8 and Figure 9. The utilization of high-grade plant resources was reported, and the relative content of each component was determined using the peak normalization method [37].

Figure 10 shows the distribution of all pyrolysis products for the four groups of samples. Biogas and bio-oil are prepared by cracking the sample. According to the Py-GC/MS results for B1, 77 thermal cracking products were detected. The corresponding percentage of each thermal cracking product was the mass obtained by cracking 100 g of raw materials. Among them, for each 100 g of B1 sample, the thermal cracking products were obtained: Ala-Gly (16.15 g), 2-Butene (10.42 g), ethyl-Cyclopropane (6.79 g), 3-iodo-1-HPyrazole (4.51 g), Furfural (3.61 g), Methyl formate (2.86 g), (E)-2-Butenal (2.36 g), 2,5-dimethyl-Furan (1.9 g), Toluene (1.24 g), Adipic acid, propyl transhex3-enylester (1.05 g) and Lidocaine (1.03 g).

According to the Py-GC/MS results for B2, 79 thermal cracking products were detected. For every 100 g of B2 sample lysed could obtain: 2-Propenal (20 g), 2-Butene (12.52 g), 2-methyl-Furan (9.15 g), Ethyl-Cyclopropane (7.71 g), 3-iodo-1-HPyrazole (5.32 g), Furfural (3.55 g), (E)-2-Butenal (2.38 g), 2,5-dimethyl-Furan (2.32 g), Methyl formate (2.26 g), Toluene (1.71 g), Benzene (1.49 g), and 1-methyl-1,3-Cyclopentadiene (0.99 g).

According to the Py-GC/MS results for B3, 69 thermal cracking products were detected. For every 100 g of B3 sample lysed, we could obtain 2-methyl-3-Buten-1ol (20.51 g), 2-Butene (14.7 g), 3methyl-Furan (7.64 g), 1,2-dimethyl-Cyclopropane (7.13 g), 1H-Pyrazole (5.62 g), Benzene (4.24 g), 2-Pentenedioic acid (2.47 g), Furfural (2.16 g), (E) 2-Butenal (2.13 g), 2,5-dimethyl- Furan (2.11%), Dodecanoic acid (1.69 g), 2-Propenoic acid (0.74 g), and Adipic acid (0.85 g).

According to the Py-GC/MS results for B4, 76 thermal cracking products were detected. For every 100 g of B4 sample lysed, we could obtain: 3-Buten-1-ol,2-methyl (18.43 g), 2-Butene (15.15 g), ethyl-Cyclopropane (7.9 g), 3methyl-Furan (6.94%), 3-iodo-1H-Pyrazole (5.23 g), Furfural (4.47%), Benzene (3.43 g), Toluene (2.74 g), Urea (2.47 g), 2,3dihydro-Furan (2.45 g), and 2,5-dimethyl-Furan (2.13 g).

The distribution of the four group samples according to their retention times is shown in Figure 11. Overall, the four groups of samples had the highest content during 5–10 min, all of which exceeded 70%. Following 10–15 min, the content was about 10–15%. After more than 15 min, the content did not exceed 5%. This indicates that during the pyrolysis process, most of the organic molecules were detected within 15 min. Compared to the four groups of samples, B1 accounted for 70.59% of the small molecules during the 5–10 min stage. The B2, B3, and B4 samples with nano-catalysts accounted for 82.9%, 82.92%, and 85.11%, respectively, during the 5–10 min stage. The results showed that the nano-catalysts Mo and Fe_2_O_3_ accelerated the decomposition of small molecules before 10 min. 

The presence of Cyclopropane, o-xylene, and 2-Butene demonstrates the presence of aromatic compounds in CCS wood [38,39]. Biomass can be converted into biofuel through pyrolysis or gasification by introducing a nano-catalyst as seen in Figure 12. Without a catalyst, acetic acid and hydroxyacetone are mainly generated, accompanied by a small amount of ketones. When the catalyst is added to the reaction process, the content of acid and ketone decreases, and the content of phenols and aromatic hydrocarbons increases, indicating that the catalyst enhances the deoxidation and aromatization reaction [40,41]. The presence of a pleasant scent of methyl-formate indicates that CCS wood exudes a particular aroma gas. In addition, it has been detected that Urea plays an important role in agriculture and industry. In agriculture, Urea is mainly used as fertilizer and animal feed. In industrial production, Urea is an important raw material for urea-formaldehyde resins [42]. Another organic substance, Lidocaine, is a local anesthetic that blocks nerve excitation and conduction by inhibiting the sodium ion channels of nerve cell membranes. It has been widely used in cosmetic plastic surgery for local infiltration anesthesia [43]. In addition, the presence of Furfural plays a role in the sterilization of CCS wood. Furfural is widely used in the pharmaceutical, pesticide, veterinary, and food industries [44,45].

Chemical Composition of CCS wood extracts

### 3.4. FT-IR Analysis

The infrared spectra of CCS wood were analyzed according to the FT-IR analysis. As shown in Figure 13, the absorption peaks of the CCS wood extract were mainly concentrated in the 3550–3300 cm^−1^, 1750–1450 cm^−1^, and 1180–680 cm^−1^ bands. The absorption peaks in the band were 3400 cm^−1^ or a higher infrared spectrum was formed by the vibration or reflux stretch of the hydroxyl group in liquid water. The absorption peak at 3500–3350 cm^−1^ was formed due to the intermolecular association near the broad peak. The absorption peak by the stretching vibration of a saturated C–H bond was formed at 3023 cm^−1^. The absorption peak at cm^−1^ was formed by the antisymmetric stretching of CH_2_ group. The absorption peak at 1750–1550 cm^−1^ was caused by the vibration of C=O double bond stretching vibration formation [46]. The absorption peak at 1450 cm^−1^ may indicate the formation of an asymmetric corner vibration of CH_3_. The absorption peak at 1181 cm^−1^ was the anti-symmetric stretching of the C–O–C of the ester substance [47]. The absorption peak at 940 cm^−1^ may be formed by the out-of-plane bending of C–OH [48]. The observed cellulose absorption peak (3550–3300 cm^−1^) decreased significantly, indicating that cellulose was hydrolyzed. The absorption peaks of lignin (1480–1430 cm^−1^) and hemicellulose (3000 and 2860 cm^−1^) decreased slightly, indicating that both hemicellulose and lignin were less hydrolyzed than cellulose [49]. The absorption peaks of the extracts were mainly concentrated in the bands of 3700–3000 cm^−1^, 1750–1500 cm^−1^, and 1100–700 cm^−1^. After the analysis, the main chemical components were terpenoids, phenols, acids, ketones, esters, and aromatic compounds. In addition, the decrease in their characteristic absorption peaks indicated that these chemical components were partially extracted.

### 3.5. GC-MS Analysis

The total ion chromatograms of the CCS wood analyzed by GC-MS are shown in Figure 14, and the specific results are shown in Appendix A. The sample was extracted to obtain liquid biogas and bio-oil. The types and contents of liquid biogas and bio-oil in the sample extract were detected by GC-MS. According to the GC-MS test results, 33 peaks were detected in A1. Among them, using an organic solvent to extract 100 g of CCS wood can get: N,N-diethyl-Formamide (0.35 g), 2-Naphthalenemethanol (0.32 g), 2-ethyl-1-Hexanol (0.25 g), 1,2,3-Benzenetriol (0.12 g), Aristol-1(10)-en-9-ol (0.04 g), Oleic Acid (0.03 g), 6-Hydroxybenzofuran-3-one (0.02 g), Resorcinol (0.02 g), Levodopa (0.02 g), and Butanoic acid (0.01 g). 

A variety of organic materials are shown in the Appendix A. For example, Formamide was an oily liquid with active reactivity and special solvency. It is often used to synthesize organic raw materials and to determine the content of amino acids in rice. For organic synthesis, many are used in medicine, as well as for pesticides, dyes, pigments, perfumes, and auxiliaries [50]. In addition, 1-Hexanol microstrip wine, fruity, and fatty flavors are commonly used in the food industry for fragrance bases and formulated essential oils, which indicate the special aroma of sassafras. Levodopa was the drug used for the treatment of tremor paralysis. It can enter the brain through the blood-brain barrier and can be enzymatically converted into dopamine, and also used to treat Parkinson’s syndrome [51,52]. In addition, 9,12-Octadecadienoic acid (Z,Z)- and Oleic Acid also played indispensable roles in the industrial and pharmaceutical industries. 9,12-Octadecadienoic acid (Z,Z)- is a nutrient and fatty acid that was used in human and animal nutrition and can combine with cholesterol to form an ester, which in turn can promote the degradation of cholesterol to bile acid, and then be excreted from the body, further reducing triglyceride effects [53,54]. Furthermore, it can be used as a raw material for the prevention and treatment of atherosclerosis. Oleic Acid can be used to produce epoxy oleate via epoxidation and as a good solvent for fatty acids and oil-soluble substances. It was also used for the precision polishing of precious metals, such as gold, silver, and non-metals [55].

## 4. Conclusions

In this study, the TGA-DTG and TG-FTIR tests showed that in the process of thermal decomposition of CCS wood, thermal weight loss was generally divided into three stages, and the pyrolysis gasification process of bio-oil was mainly divided into two stages: the volatilization pyrolysis of light components under low temperature conditions and the pyrolysis gasification and condensation coking of heavy components under high temperature conditions. In addition, the absorption peaks of the stretching vibration of the C-H bond, the C=O double bond stretching vibration, and the C-H bond bending vibration in the plane correspond to different macromolecular substances, such as ketones, aldehydes, alcohols, ethers, and acids. Comparing the four sets of data, the metal nano-Mo catalyzes the generation of gaseous substances, and the absorbance of gaseous substances is significantly higher than that of catalyst-free logs. The Py-GC/MS test detected more than 50 pyrolysis products in each sample, mainly hydrocarbons, alcohols, and phenols, which are beneficial to biomass oils. The absorption peaks of cellulose (3550–3300 cm^−1^), hemicellulose (3000 and 2860 cm^−1^), and lignin (1480–1430 cm^−1^) decreased slightly, indicating cellulose, hemicellulose, and wood, according to FTIR analysis. The GC-MS test detected more than 30 peaks. Among the tested substances, 1-Hexanol microstrip wine, fruity, and fatty flavors are commonly used in the industry for perfume bases and formulated essential oils, which explain the particular aroma of yellow ash. 

Nano-catalysts Mo and Fe_2_O_3_ affected the composition of aromatic compounds, acids, and alkanes. The samples containing nano-Mo showed a good catalytic effect, which indicated that nano-Mo greatly improved the pyrolysis rate under the catalysis of the metal catalyst. According to the characteristics and characteristics of biomass pyrolysis technology, CCS wood was used in combination with a nano-metal catalyst for the first time, which is conducive to the development of CCS wood into high value-added products and provides a basis for the comprehensive utilization of high-quality resources. 

## Figures and Tables

**Figure 1 polymers-14-04610-f001:**
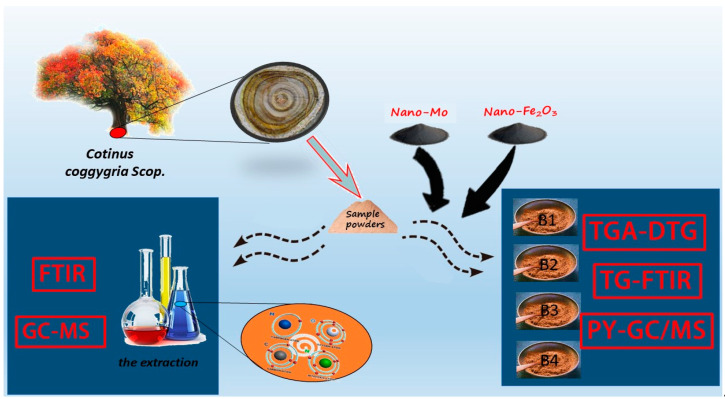
The flow chart of CCS wood.

**Figure 2 polymers-14-04610-f002:**
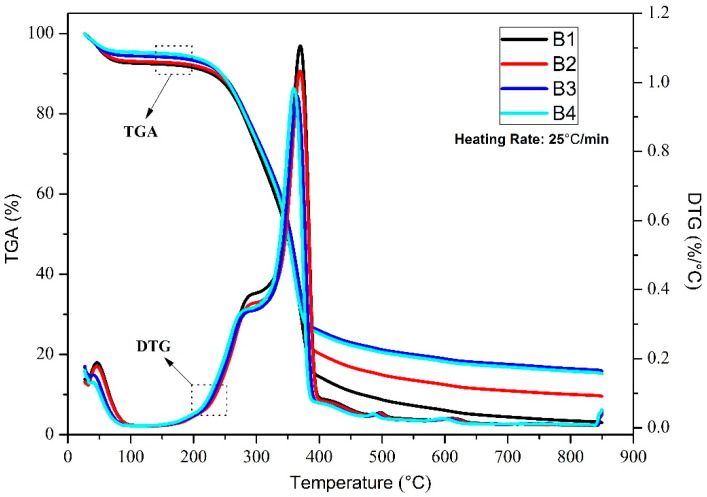
TG and DTG curves of B1, B2, B3, and B4 at 25 °C/min.

**Figure 3 polymers-14-04610-f003:**
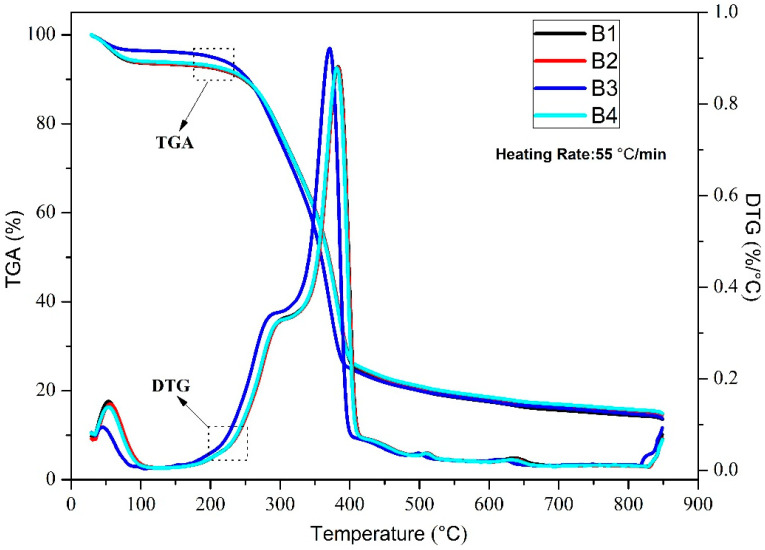
TG and DTG curves of B1, B2, B3, and B4 at 55 °C/min.

**Figure 4 polymers-14-04610-f004:**
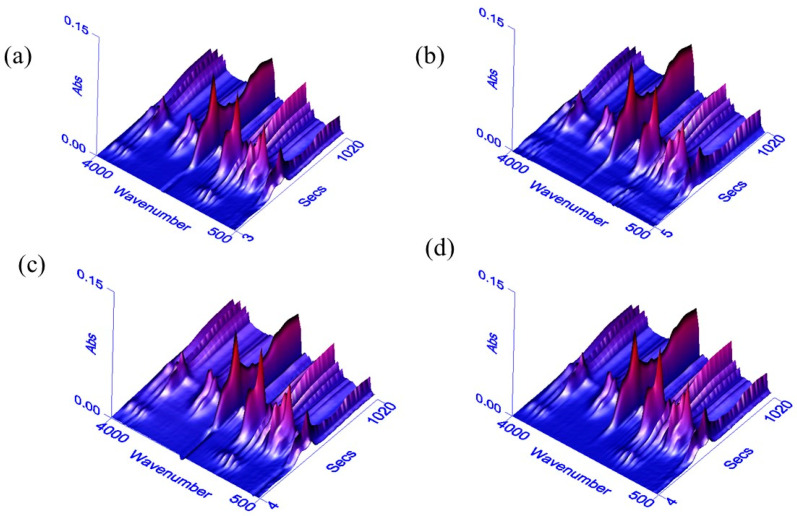
3D FTIR spectrograms of the samples: B1 (**a**), B2 (**b**), B3 (**c**), and B4 (**d**).

**Figure 5 polymers-14-04610-f005:**
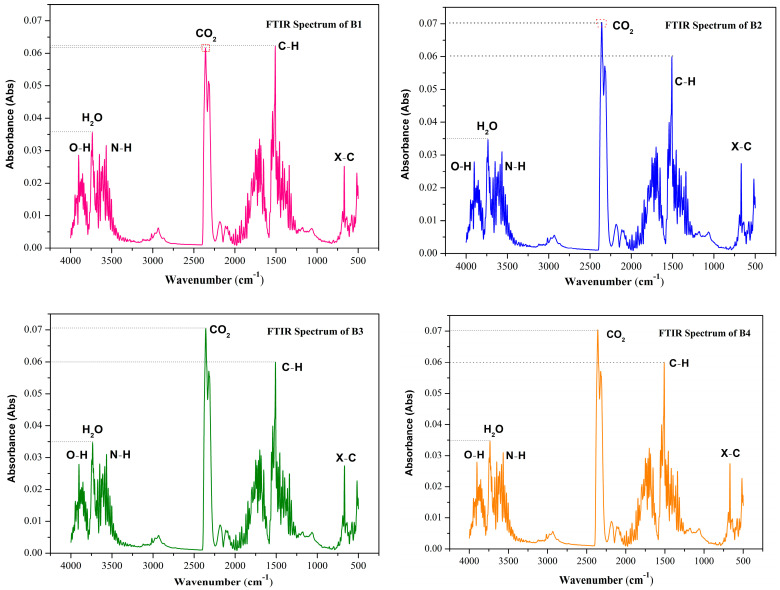
FTIR spectrum of pyrolysis products from **B1**, **B2**, **B3**, and **B4**.

**Figure 6 polymers-14-04610-f006:**
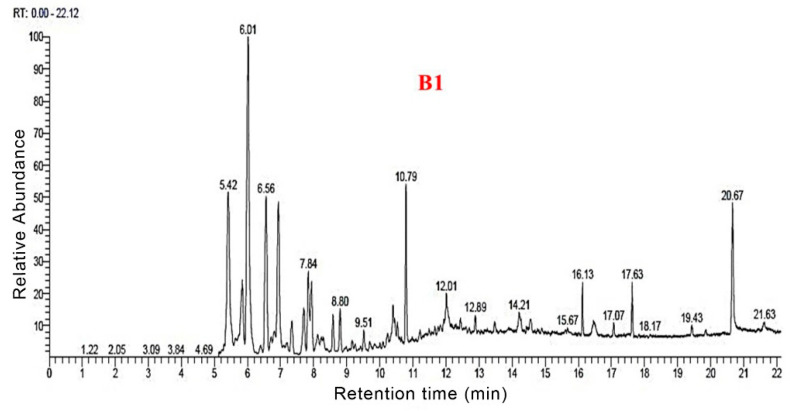
Total ion flow map of B1 by Py-GC/MS.

**Figure 7 polymers-14-04610-f007:**
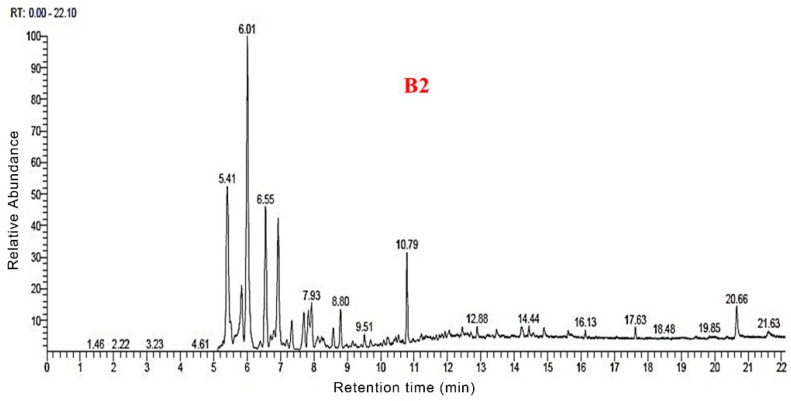
Total ion flow map of B2 by Py-GC/MS.

**Figure 8 polymers-14-04610-f008:**
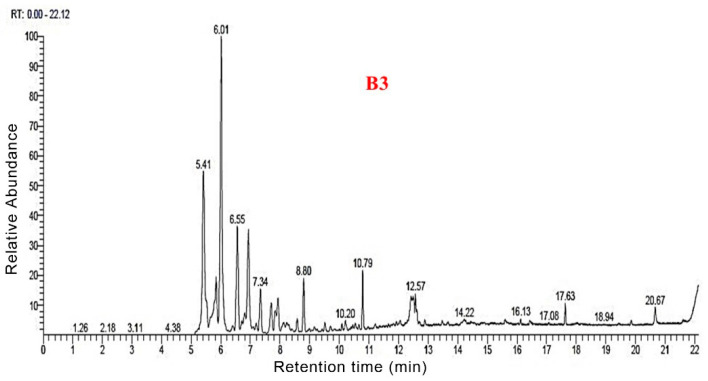
Total ion flow map of B3 by Py-GC/MS.

**Figure 9 polymers-14-04610-f009:**
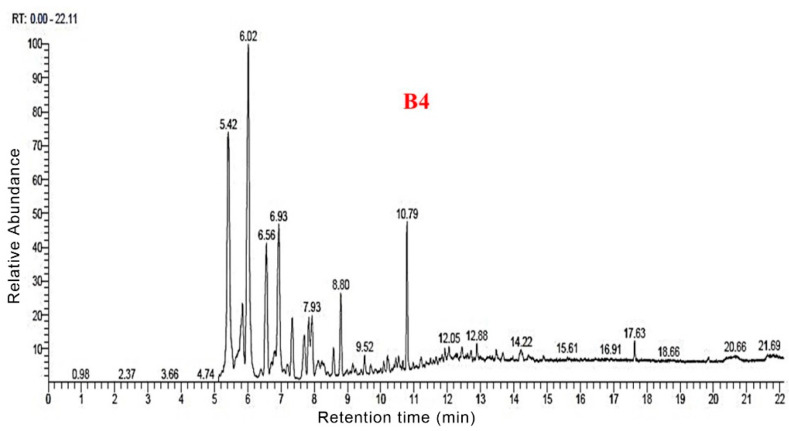
Total ion flow map of B4 by Py-GC/MS.

**Figure 10 polymers-14-04610-f010:**
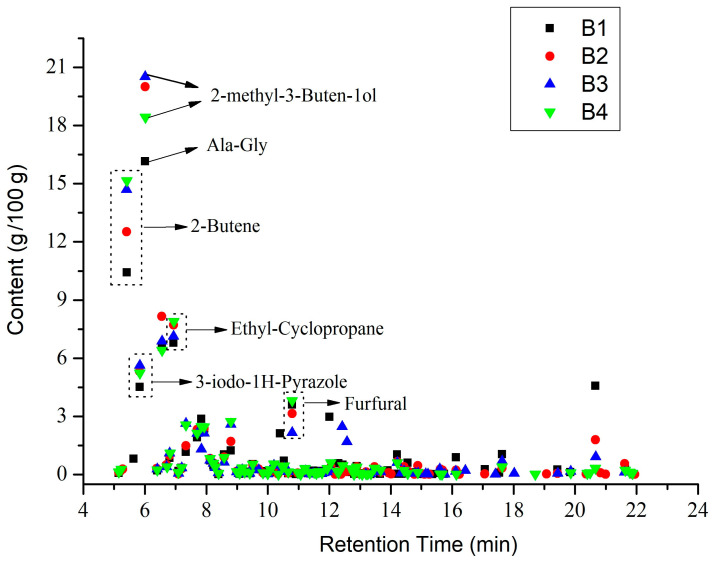
Pyrolysis product distribution diagram by Py-GC/MS.

**Figure 11 polymers-14-04610-f011:**
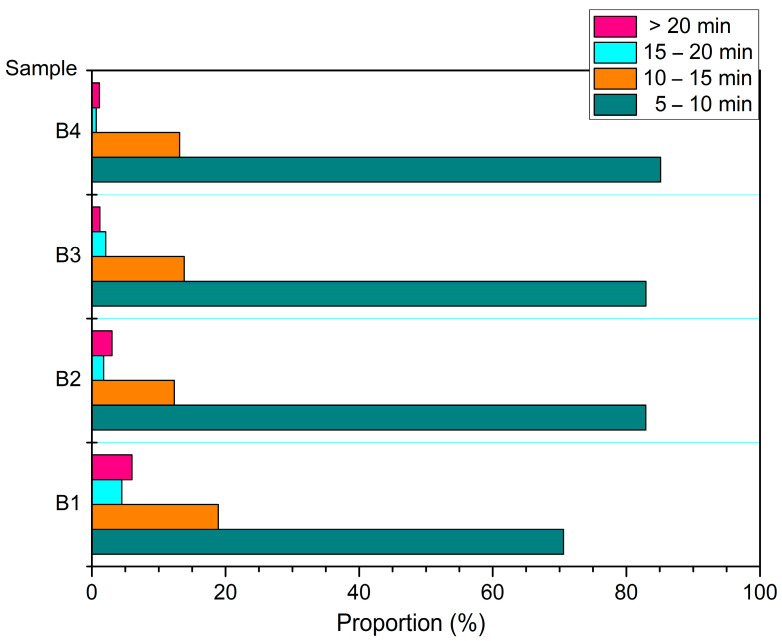
Distribution characteristics of the sample according to retention time.

**Figure 12 polymers-14-04610-f012:**
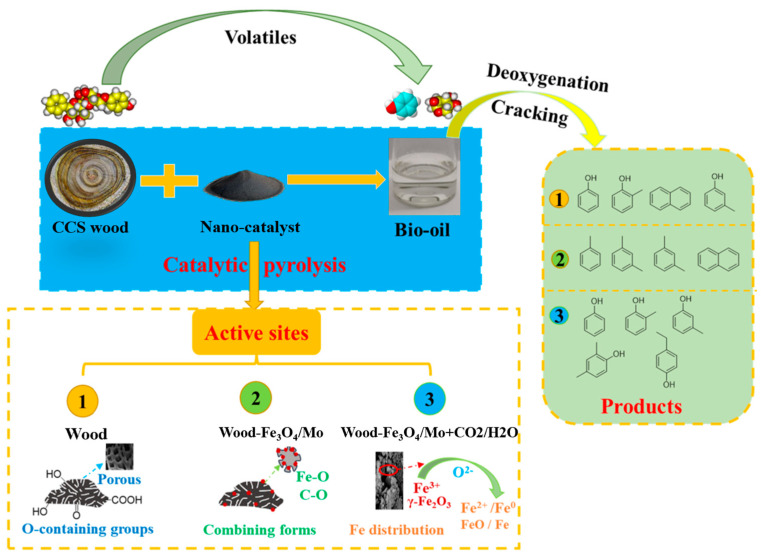
Reaction mechanism of the nano catalysis experiment.

**Figure 13 polymers-14-04610-f013:**
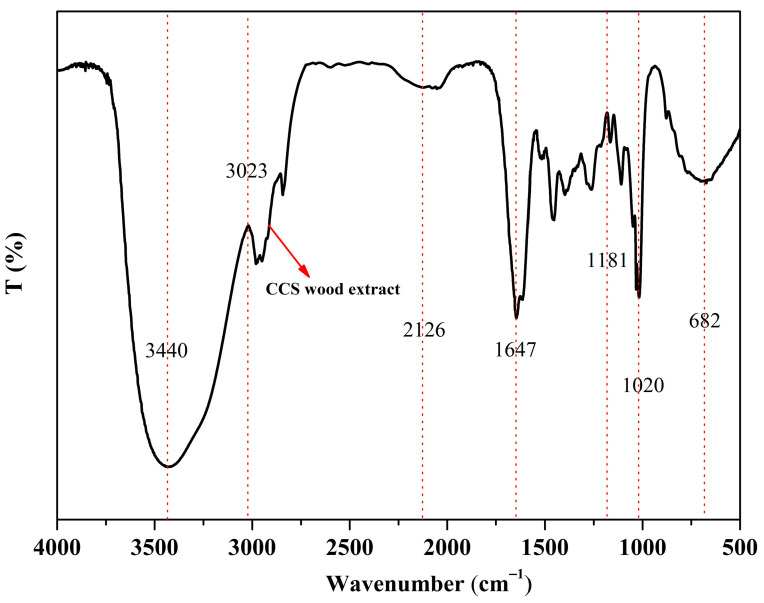
FT-IR spectra of CCS wood extract.

**Figure 14 polymers-14-04610-f014:**
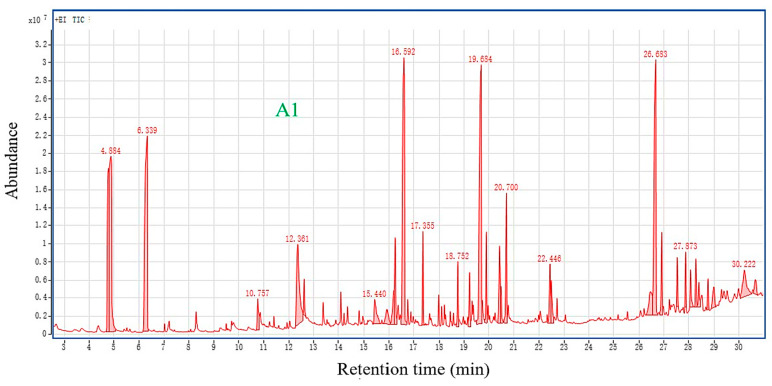
Total ion chromatograms of the extract of CCS wood.

**Table 1 polymers-14-04610-t001:** Optimization of the experimental program.

No.	Material	Catalyst
B1	Wood sample (5 g)	-
B2	Wood sample (5 g)	Mo (0.05 g)
B3	Wood sample (5 g)	Fe_2_O_3_ (0.05 g)
B4	Wood sample (5 g)	Mo (0.025 g)/Fe_2_O_3_ (0.025 g)

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
