# Peer review of "Nano Catalysis of Biofuels and Biochemicals from *Cotinus coggygria* Scop. Wood for Bio-Oil Raw Material"

_polymers, 2022, doi:10.3390/polym14214610_

Round 1
Reviewer 1 Report
The research topics and methodology used are very interesting. The obtained results are well described and discussed with other works. This manuscript is almost ready and requires minor revisions before it is ready for publication. Further detailed comments for consideration are provided below.
Comment 1#
Authors should use the new Polymers Microsoft Word template file (2022).
Comment 2#
Lines 36, 38, 39, 50: "et al." instead of "et al"
Comment 3#
Line 63: remove “And”
Comment 4#
Line 311: “acids” instead of “Acids”
Comment 5#
Correct references according to journal guidelines.
Author Response
Comments from Reviewers
Reviewer #1:
The research topics and methodology used are very interesting. The obtained results are well described and discussed with other works. This manuscript is almost ready and requires minor revisions before it is ready for publication. Further detailed comments for consideration are provided below.
Answer: Thanks a lot for your compliments and feedback to our manuscript.
Q1. Authors should use the new Polymers Microsoft Word template file (2022).
Answer: We thank the reviewer very much for the comments. We have used the latest Microsoft Word template file.
Q2. Lines 36, 38, 39, 50: "et al." instead of "et al".
Answer: Thank you very much for your comment, "et al" have been revised.
Q3. Line 63: remove “And”.
Answer: Thank you very much for your comment, this sentence has been revised.
Q4. Line 311: “acids” instead of “Acids”
Answer: Thank you very much for your comment, “Acids” has been revised to “acids”.
Q5. Correct references according to journal guidelines.
Answer: Thank you very much for your comment. All references have been revised according to the journal format.

Reviewer 2 Report
Dear editor, dear author,
The work of Yue et al. tries to understand the chemistry of the extract of one specific shrub and to observe the effect of a couple of catalysts. However, is overall unclear the extraction (?) procedure and hence the nature of the powders they threat in the article.
Underneath I summarize all the major concern related to this manuscript:
Introduction is limited, the plant effects are somehow described, but the techniques applied is not present, no related previous similar works on other plants… This part must be implemented.
Table 1 has no significance. The authors only studied 1 solvent mixture.
It’s unclear what’s the authors did with the catalyst. No details on extraction are given. Temperature? Time? Proportions? In figure 1, it’s also not clear what the author did with the catalyst… It looks just like a solid addition. Please improve.
The further analysis was done on the powders? On the extract? On the exhausted? Unclear.
Being this part unclear, also the interpretation of the other parts is hard to understand in the result & discussion section.
TGA analysis has nothing to do with flame retardancy (line 132). Please reconsider.
Figure 4 is wonderful, but no difference can be seen between the 4 experiments. Please elaborate so that the main difference can be highlighted. Same is for figure 5, the information acquired are really overlapping.
Also in Fig.9, it would be important that the authors put a main focus on the main differences between extracts (?).
In relation to the drawn conclusion, is hard to agree or disagree being unclear what is the subject of the analysis
Further, not less important question to be addressed is related to the inherence of the topic to the journal scope. In my opinion the interest for this particular journal is limited.
According to this observation I would suggest the authors to resubmit the paper once it will be deeply restructured.
Author Response
Reviewer #2:
The work of Yue et al. tries to understand the chemistry of the extract of one specific shrub and to observe the effect of a couple of catalysts. However, is overall unclear the extraction procedure and hence the nature of the powders they threat in the article.
Underneath I summarize all the major concern related to this manuscript:
Q1. Introduction is limited, the plant effects are somehow described, but the techniques applied is not present, no related previous similar works on other plants… This part must be implemented.
Answer: We thank you for your careful review and for given us a possibility to improve the quality of our manuscript. We added some similar studies on other plants in the introduction.
Action: “Ge et al. (2022) studied the potential value of Pinus armandii by using TG, PY-GC-MS, FTIR, NMR and other technologies, and found that Pinus armandii was rich in natural product active molecules and bio-oil, which can be used for the development and uti-lization of the biomedical industry [13]. Ma et al. (2022) used FTIR, GCMS, LC-QTOF-MS and other methods to analyze the bioactive components of Ginkgo biloba branches, and realized the zero waste multi-level utilization of biomass of Ginkgo biloba branches [14]. The scavenging capacity of Ginkgo biloba branches' bioactive compounds to free radicals reached 98.18%.”
Q2. Table 1 has no significance. The authors only studied 1 solvent mixture.
Answer: We thank the reviewer very much for the comments. Table 1 has been deleted.
Q3: It’s unclear what’s the authors did with the catalyst. No details on extraction are given. Temperature? Time? Proportions? In figure 1, it’s also not clear what the author did with the catalyst… It looks just like a solid addition. Please improve.
Answer: We thank the reviewer for the comments. We have modified the Experimental materials according to your comments.
Q4: The further analysis was done on the powders? On the extract? On the exhausted? Unclear. Being this part unclear, also the interpretation of the other parts is hard to understand in the result & discussion section.
Answer: We thank the reviewer very much for this question. In 3.1-3.3, we studied the behavior of CCS wood powder combustion process after adding catalyst, and in 3.4 and 3.5, we studied the chemical composition of CCS wood extracts. We have marked two directions in the text. “Behavior during Combustion of CCS wood, Chemical Composition of CCS wood extracts”.
Q5. TGA analysis has nothing to do with flame retardancy (line 132). Please reconsider.
Answer: We thank the reviewer very much for the comments. we have revised this description. “The burning characteristics and moisture content of wood determine whether wood has excellent flame retardancy.”
Q6. Figure 4 is wonderful, but no difference can be seen between the 4 experiments. Please elaborate so that the main difference can be highlighted. Same is for figure 5, the information acquired are really overlapping.
Answer: We thank the reviewer very much for the comments. Fig. 4 shows the 3D-FTIR spectral structure of the pyrolytic volatiles from the time axis and absorption peak, and Fig. 5 shows the intensity of each absorption peak. Figure 5 has been modified to show that the content of small molecular products such as CO2 increases under the action of catalyst.
Q7. Also in Fig.9, it would be important that the authors put a main focus on the main differences between extracts (?).
Answer: We thank the reviewer for the comments. Figures 6-9 shows the ion current spectrum of small molecule products detected during pyrolysis. The content and cracking time of products under the action of catalyst were different. The difference of content has been given in the text, and the difference of cracking time was shown in Fig. 10.
Q8. In relation to the drawn conclusion, is hard to agree or disagree being unclear what is the subject of the analysis. Further, not less important question to be addressed is related to the inherence of the topic to the journal scope.
Answer: We thank the reviewer very much for the comments. We have made changes in the manuscript. Our main purpose is to use the characteristics of biomass pyrolysis technology and the advantages of nano catalysts to develop and create high value-added products for CCS wood, so as to provide a basis for the comprehensive utilization of high-quality resources.

Reviewer 3 Report
The article submitted for review proves the various uses of Cotinus coggygria Scop .. Cotinus coggygria Scop. as a precious landscape shrub and a good afforestation species this is used in the pharmaceutical industr. The authors of the article proved that Cotinus coggygria is also an excellent biomaterial for biofuels and biochemicals. In the introduction, the possibilities of use are presented in a very broad way of Cotinus coggygria Scop.. It is a very good publication. All research procedures used were described in detail. There was a good discussion and analysis of the obtained research results.
However, I have a few comments:
1. In Figure 5, when describing axes, please use a space (before the unit);
2. In figures 6-9, please correct the axis descriptions so that they are visible and legible;
3. In the descriptions of axes in figures 12 and 13, please use a space;
4. In chapter 2.2, please explain the abbreviations (all) the first time you use it.
Moreover, the publication is a valuable source of information and forms the basis for further research.
Thank you for considering my opinion. I encourage the authors to continue working on improving the manuscript.
Author Response
Reviewer #3:
The article submitted for review proves the various uses of Cotinus coggygria Scop. Cotinus coggygria Scop. as a precious landscape shrub and a good afforestation species this is used in the pharmaceutical industry. The authors of the article proved that Cotinus coggygria is also an excellent biomaterial for biofuels and biochemicals. In the introduction, the possibilities of use are presented in a very broad way of Cotinus coggygria Scop. It is a very good publication. All research procedures used were described in detail. There was a good discussion and analysis of the obtained research results. However, I have a few comments:
Q1. In Figure 5, when describing axes, please use a space (before the unit).
Answer: We thank the reviewer very much for the comments. We have revised the Figure 5.
Q2. In figures 6-9, please correct the axis descriptions so that they are visible and legible.
Answer: We thank the reviewer very much for this question. We have revised the Figures 6-9.
Q3. In the descriptions of axes in figures 12 and 13, please use a space.
Answer: We thank the reviewer very much for the comments. We have revised the Figures 12 and 13.
Q4. In chapter 2.2, please explain the abbreviations (all) the first time you use it.
Answer: We thank the reviewer very much for the comments. All the first abbreviations have been explained.

Reviewer 4 Report
This research studied about Nano Catalysis of Biofuels and Biochemicals from Cotinus coggygria Scop. Wood for Bio-oil Raw Material. After reviewing it carefully, I think the overall of this manuscript is good and interesting for reader. However, some revision should be considered before considering on publication. My specific comments are as follows:
- When I see the title in this manuscript, it looks like the catalyst with catalytic mechanisms will be focused on this manuscript. But the fact that no these information are provided, especially for catalyst characterization part. Please provide it carefully.
- nano-Mo/Fe2O3 is applied. Is it the best one? How about other nano metal?
- I suggest author do more experiment about catalyst reusability test. It is very important to let people know the possibility to be further applied in practical process.
- Please provide one figure about catalytic reaction partway including their discussion.
Author Response
Reviewer #4:
This research studied about Nano Catalysis of Biofuels and Biochemicals from Cotinus coggygria Scop. Wood for Bio-oil Raw Material. After reviewing it carefully, I think the overall of this manuscript is good and interesting for reader. However, some revision should be considered before considering on publication. My specific comments are as follows:
Q1. When I see the title in this manuscript, it looks like the catalyst with catalytic mechanisms will be focused on this manuscript. But the fact that no these information are provided, especially for catalyst characterization part. Please provide it carefully.
Answer: We thank the reviewer very much for the comments. In this manuscript, we focus on developing CCS wood into a high value-added product, using nano catalysts to affect the composition of aromatic compounds, acids and alkanes, and improve the pyrolysis efficiency of CCS wood, thus providing a basis for the comprehensive utilization of CCS wood.
Q2. nano-Mo/Fe2O3 is applied. Is it the best one? How about other nano metal?
Answer: We thank the reviewer very much for the comments. In the section of Py-gc/ms analysis, “…This indicated that during the pyrolysis process, most of the organic molecules have been detected within 15 min. Compared to the four groups of samples, B1 accounted for 70.59% of the small molecules during the 5-10 min stage. The B2, B3, and B4 samples with nano catalysts accounted for 82.9%, 82.92%, and 85.11%, respectively, during the 5-10 min stage.” Nano-Mo/Fe2O3 (B4) obtained the most small molecule products in the first 10 minutes, and the catalytic efficiency is the best.
Q3. I suggest author do more experiment about catalyst reusability test. It is very important to let people know the possibility to be further applied in practical process.
Answer: We thank the reviewer very much for the kindly advise. In the subsequent experiments, we will conduct more perfect experiments on the nano-catalytic pyrolysis experiment, so as to understand the mechanism of catalytic reaction and select a more appropriate combination of nano catalysts.
Q4. Please provide one figure about catalytic reaction partway including their discussion.
Answer: We thank the reviewer very much for the comments. We added some discussion in 3.3. Py GC/MS analysis.
The presence of Cyclopropane, o-xylene, and 2-Butene demonstrates the presence of aromatic compounds in CCS wood. Biomass can be converted into biofuel through pyrolysis or gasification by introducing nano catalyst as seen in Fig. 12. Without catalyst, acetic acid and hydroxyacetone are mainly generated, accompanied by a small amount of ketones. When the catalyst is added to the reaction process, the content of acid and ketone decreases, and the content of phenols and aromatic hy-drocarbons increases, indicating that the catalyst enhances the deoxidation and aro-matization reaction [40, 41].

Round 2
Reviewer 2 Report
Dear editor, dear authors,
Now the experimental part is clearer, but the interest of the article is limited.
Indeed, the authors added some metal catalyst (1%) to a wood powder and made analysis on these powders which are very similar to each other (of course). Especially TG, FT-IR experiments can be overlapped.
Py-GC/MS are also not significantly different.
To me this study proves only that adding 1% of metal in wood powder, the techniques used for the characterization are not powerful enough to identify the differences.
The analysis of the extract (A1) is a dissociated part of the study that could be considered separately. The findings of this extract are not compared with anything else so they are, as such, not interesting.
On the bases of these observation I firmly confirm the rejection of this study.